# Current Trends and Changes in Use of Membrane Molecular Dynamics Simulations within Academia and the Pharmaceutical Industry

**DOI:** 10.3390/membranes13020148

**Published:** 2023-01-24

**Authors:** Stephan L. Watkins

**Affiliations:** Plant Pathology and CRGB, Oregon State University, 2701 SW Campus Way, Corvallis, OR 97331, USA; stephenlloydriggs@gmail.com

**Keywords:** molecular dynamics (MD), sphingolipid, eicosanoid, ceramide, lipid rafts, vesicle, graphics processing unit (GPU), biologics, petaflop, drug design, drug delivery, rational design, PC, phosphatidylcholine, PS, Phosphatidylserine

## Abstract

There has been an almost exponential increase in the use of molecular dynamics simulations in basic research and industry over the last 5 years, with almost a doubling in the number of publications each year. Many of these are focused on neurological membranes, and biological membranes in general, applied to the medical industry. A smaller portion have utilized membrane simulations to answer more basic questions related to the function of specific proteins, chemicals or biological processes. This review covers some newer studies, alongside studies from the last two decades, to determine changes in the field. Some of these are basic, while others are more profound, such as multi-component embedded membrane machinery. It is clear that many facets of the discipline remain the same, while the focus on and uses of the technology are broadening in scope and utilization as a general research tool. Analysis of recent literature provides an overview of the current methodologies, covers some of the recent trends or advances and tries to make predictions of the overall path membrane molecular dynamics will follow in the coming years. In general, the overview presented is geared towards the general scientific community, who may wish to introduce the use of these methodologies in light of these changes, making molecular dynamic simulations more feasible for general scientific or medical research.

## 1. Introduction

With the advent of higher computing power over the last several years, there has been a substantial increase in the use of molecular dynamics simulations (MD) in various studies incorporating membranes [1,2] (Figure 1). These range from simple single or di-lipid mixtures to complex biological systems including proteins, glycolipids, sphingolipids, ceramides or several specialized compounds and cellular membrane components of interest [3,4]. In many, there have also been attempts to incorporate complex structures from bilayers and double bilayer systems, irregular attached layers in engineering and lipid monolayers on a range of surfaces to vesicles used in research and medicine [5,6,7]. Based on current trends in computer speed and costs, petaflop computer systems are now in the price range of single labs or small departments. This indicates that conducting properly designed research experiments across disciplines is increasingly important, as the use of simulations for varied aspects will only increase in the coming years [8,9]. In biology, the rate for design of novel macromolecular cures has increased substantially, with computational sciences as the main driving force. These trends indicate a major change in pharmaceutical sciences from small molecule to complex biologics, which cure rather than treat most medical issues, and invariably change the overall focus from industry and the industrial support given by academic biology in general. As the primary fields of biological research move away from pharmaceutically driven focuses, an increase in basic biological studies will probably ensue in several years’ time. Many of the MD techniques used in medical research translate well to more dynamic research, such as plant cell membranes, organelles and a diversity of organisms’ cellular membranes. There are currently a wide range of reviews which discuss the techniques involved, and various software available for researchers [10,11,12,13]. Many of these reviews also discuss proper model generation, membrane construction and evaluation procedures. These techniques and models will be mentioned with appropriate references; however, the focus of this review is on trend changes and incorporating membrane structural work into general research laboratories and biologics development. Additionally, this review should allow scientists unfamiliar with MD to critique the increasing number of published research articles in this subdiscipline.

From the prospect of information, the power of MD cannot be disputed, limited only by the design of the initial model systems involved. This initial design process incorporates the complexity of the system involved, related to molecular components, and the types of simulation. Overall, MD have been grouped into four broadly defined criteria. Types of simulations range from coarse grained (CG), to all atom, to all atom including quantum mechanics. There is a direct trade off between the detailed modeled molecular representations and computational load, or time, necessary to conduct an experiment. General descriptions of the four types of models include CG, made of hybrid systems with several atoms combined into a single point, hybrid systems with only some aspects combined, partial atom, which only fuse non-polar hydrogen into a single point, and all atom, which are self-explanatory. Additionally, there are models which incorporate quantum mechanics, allowing electron transfer and covalent bond changes to be included. These currently are usually only utilized for a small portion of larger models in biological or complex systems due to an exponential increase in computational load, and are often only necessary for engineering or non-pharmaceutical research in chemistry and basic biology [14,15]. Additionally, most academicians are limited by the overall analysis utilized to presenting aspects of simulations based on the initial design of the simulation project and available computational resources. Often, a wealth of information remains in the raw data itself, with increasing levels based on level of detailed atomic representation incorporated into the MD. Well focused laboratories can, with extensive simulations and analysis, generate more than one publication from experiments, or deposit raw data for analysis in newly emerging repositories from several laboratories [16,17,18]. From a researcher’s perspective, combining MD experiments with wet laboratory techniques can provide a rational approach to interpretation, as well as a focus for a particular project. Basic research simulations can shed light on often difficult-to-examine processes, lipid interactions, membrane proteins or the mitigation of observations from different groups.

Within all of this, the pharmaceutical industry has developed a large amount of private MD units within its own research and development [19]. In many of these, the focus has been on small molecule based approaches, with an emerging market for designed chimeric, antibody based or engineered biologics over the last 15 years [20,21]. Among the most utilized aspects of MD are drug targets incorporating membrane proteins. These include receptors, such as CD19 and CD20, with regulatory effects in cells, to neurologically based receptors effecting synaptic transmission, and complex machinery such as ATP Synthase or Rhodopsin [22]. Much of the early utilized biologics design approaches only relied on static structures, such as superposed models, to generate chimeric anti-CD19 in CAR-T cells. With a growing wealth of domain function analysis, and understanding of the molecular process in its dynamic aspect, better approaches can be followed [23]. Further, many products can be tested for proper functioning before moving to a real-life synthetic phase. The overall comparisons between atomistic or partial hybrid MD and wet lab show an extremely high level of homology, from kinetics to expected cellular or other function. In most cases where discrepancies exist, there was a lack of proper model creation in the MD itself, or problems in wet laboratory experiments [24,25]. Here, MD often serve as another form of cross checking of laboratory data. On the MD side, the most common discrepancy with kinetics arises from the lack of the repeating of simulations enough times from different initial conformations to determine proper means. Overall, the effects of entropy, conformational states at the molecular level, can give a wide range of different kinetics for a single steered simulation which do not occur in wet laboratory kinetic experiments measuring millions of simultaneous reactions. Thus these types of MD for kinetics require multiple starting points, outlined in the design process of most of the software manuals, to accurately reflect wet laboratory findings.

Another changing aspect is that most initial studies were limited in computer power. Often laboratories only looked at portions of receptors of interest with small membranes of only 40–150 lipids, with key factors related to structural integrity and function outside of a small domain unincorporated into the research. Further limits were presented from a need to conduct multiple simulations from different initial conformational states for kinetics and proper dynamic analysis, as these translate into computational resource needs. Based on mathematical accuracy at the simulation level, referred to as single or double precision, 60–100 single or 10–20 double precision simulations were required. In computer terms, single and double precision is the decimal place used by the mathematics portion of any software, with single truncated to two and double to the tenth or more decimal. In simulations, these accumulate any inaccuracy every time step calculation for forces for every atom in the system at the set decimal level. These two factors again translate to computational load. Currently however, with the use of graphics processor unit systems, a lab can conduct one single precision simulation in 24 h with a desktop computer equipped with a 16 core CPU and two graphics processors. For the corresponding double precision MD, the time involved increases to a few days. This change, however, is drastic, and prior to 2014 these same experiments could take months. Technological advancements are now moving the use of MD from a requirement for large scale clusters or supercomputers to small systems, small clusters or affordable computer resources.

Several good research examples from earlier studies were conducted incorporating membranes and proteins, or originally only a few hundred atoms. Often the feat was astounding in terms of time consumption but limited in nature, due to computer access, to a few research laboratories [26,27,28]. Some of the earliest uses of microprocessors were for the advancement of molecular simulations, even quoted as the reason why the microprocessor was invented. Currently, the trend is to incorporate ever increasing complexity to the initial model systems used. In pharmaceutical sciences, the majority of targets have been proteins embedded in cell membranes, before any computer aspects began to be used in design processes. Many receptors are dependent on unique lipid or cholesterol derived components, such as gangliosides, leukotrienes or prostaglandins, which had not been studied as cofactors or for secondary effects. In many cases, specific membrane components play vital roles in the targeted proteins, ranging from cofactors to structurally required components. Thus, the importance of the membrane structure and function cannot be underplayed, especially related to the pharmaceutical industry or medicine. Now more complex systems are beginning to be studied from smaller groups, inclusive of the entire membrane systems [29,30,31]. Primarily there was initially a divergence in the field, where membranes or proteins were simulated alone but not combined. A number of the studied membrane proteins were lacking membrane embedded or associated components. Rhodopsin is the classic model for both structural and membrane based simulations, as it was for a time the only x-ray determined membrane protein completely intact with a membrane in the initial structure [32]. As an example, Rhodopsin is dependent on several membrane lipids for structural integrity, while the majority of signaling is transmitted across bi-layers and initially even between photons and atoms [33]. However, most of the information from simulations has been accumulative, based on increases in computer power and speed, but have utilized all four simulation types over the years. Along these lines, there have now been many attempts made, with failures and successes, to incorporate membrane components for other studied proteins analogous to Rhodopsins [34].

Some important aspects related to the pharmaceutical industry include several novel technologies related to cancer or genetics. In the cancer field, membrane MD studies have greatly benefited peptide, modified peptide, cyclic peptides and variations therein. With these, most of the focus is related to either permeability or pore formation in targeted cells. This has also proved useful for the equivalent antibiotic based structures, with some of the original membrane MD studies related to small molecules’ effects on simple membranes [35,36]. There has also been some good recent research leading to design in vesicle delivery systems. While vesicle research has many aspects, MD has incorporated fusion kinetic studies, effects of incorporation of proteins for targeting, and various temperature and delivery aspects related to membrane composition and stress [27,37]. These types of study have also been included in at least one product in clinical trials related to drug delivery systems for inhaled aerosol dispersed vesicles [4,38]. Other studies have also focused on viral like engineered capsids, antibodies, chimeric antibody products interaction with membranes or membrane embedded proteins. In these cases also, MD was used for current developments in biologics in primary studies for viral like delivery systems and antibodies, and included in patent applications [39,40]. In an older focus, the membrane is looked at as something to be traversed to reach a target, transiently interacting with the target or the targeting compound itself. With some advances in membrane based hormones and various eicosanoids studies, this view is changing, and now the membrane is also seen as a place of direct interaction either as a target or cofactor. In many cases the ability to deliver a biologic, especially in the growing field of genetic manipulation, combines surface protein–membrane interactions and internal membranes such as endosomes, lysosomes or phagocytic membrane processes which change initial states [41]. Secondary membranes have only recently become more of a focus, as both secondary effects and direct targeting for designed products. Some additional aspects include off target effects of antibodies, engineered immune cells and specialized direct lipid targets such as eicosanoids. With immune cells in general, there is a large number of unknown interactions with membrane components effecting targeted cells and the engineered cells’ behaviors. Eicosanoids play extensive regulatory roles in many of the immune cell’s functions, as an example, which have often been overlooked due to the difficulty in studying the components with wet laboratory techniques in live organisms. All these factors interplay with the use of membrane MD in the design or testing phases of drugs, biologics and in general medical procedure development.

## 2. Overview of the Creation of Membrane Molecular Dynamics Simulations

After 1999, a rise in the amount of different MD simulation software was seen, mirroring in many ways the increases in computational power. This was observed again with the introduction of graphics processor unit (GPU) computing around 2008, and many of the more classic software suits established GPU versions as well. These have increased speeds for both double and single precision compiled versions of any software, with both single and double precision being feasible for a laboratory with 2–4 desktop computers equipped with several GPUs. Benchmarks for different systems and software have been published several times over the years [8,42]. In beginning any MD, the obvious first decision is which software to use and what systems or resources are available [43]. A wealth of reviews exist for this, and usually include the older original suits which are still updated regularly such as NAMD, Gromacs, and LAMMPS [14,44,45]. For higher end computations, these also incorporate auxiliary quantum mechanical (QM) software to model electron dynamics, such as GAUSSIAN, GAMESS and CP2K [46,47,48]. In general Gromacs and NAMD are usually utilized by biologists, while LAMMPS is used primarily by engineers and physicists; however, these software suits are interchangeable.

Another equally variant portion of any software is referred to as the force field (FF), which like the base software has seen a rapid rise in numbers since 2000 (Figure 2). Gromacs alone has 20 different FF types and all the software allows for introduction of user defined FF. These are not completely fields as defined in physics terms, but rather tabulated parameters such as bond lengths, angles and dihedrals alongside atomic parameters such as charge and radius. At the atomic radius level, FF definition ends with Lennard-Jones or equivalent potential energies data. Use of potential energies is a classical force field, unlike the more tabulated data, and is only used when atoms move less than an angstrom user defined distance, or close to their approximate atomic radius. Like Fortran, these parameters are used in tabulated calculations for every atom or point charge represented in the MD system defined by the user [49]. When choosing the FF, the specific questions the researcher wishes to address need to be taken into account. The four types of MD are defined by the FF utilized, which effects various results in simulations. Coarse grained MD are usually used to rapidly define macroscopic interactions such as an entire protein with a membrane over larger time periods. Hybrid systems include FF for lipid representations as entire head groups, and lipid tail represented by two single point charges per molecule. Our second set of FF are hybrid, representing each atom, with all hydrogens fused as a single point charge, and partial hybrid, where only non-polar hydrogens are fussed into a single point charge. This letter is the most commonly used and represents kinetics and conformational changes accurately enough to mirror wet laboratory findings. Final sets of FF are all atom, and all atom including QM, to model electron transfer. Currently this later type of FF has increased in use substantially, but discrepancies exist as to accuracy. Most of this accuracy is related to the QM portion which, combined with an all atom FF, tends to overestimate energies associated with electron transfer reactions. Complete QM MD seems to fit chemical derived data closely, but are mainly used for solid state and small molecule based molecular dynamics [50,51].

All atom MDs are now becoming more common due to higher computer speed for less cost and are slightly more accurate than the mentioned partial hybrid FF [52,53]. The use of all atom FFs has seen an increase in use since 2014, now making up 20–30 percent of published MD research. Equilibration time increases the more advanced the simulations representations, which can be extensive for larger simulations incorporating millions of atoms. Due to computational costs involved, QM is limited usually to only a smaller portion of the MDs where necessary, as the code adds dozens to hundreds of calculations per atom representation. Complete QM simulations of a dozen atoms can take as much time as a coarse grained simulation of a million atoms, as a direct comparison. Currently there is now a split occurring between QM simulation types with the introduction of Density Functional Theory (DFT), not to be confused with Fourier analysis. These emerging QM simulation software suits are equivalent to the hybrid MD systems for outer electron shells, allowing electron transfer to be modeled more rapidly [33,47]. The referenced software all have extensive manuals that explain force fields, MD and additional parameters in great detail, along with various computational needs and mathematics of the calculations involved. Of note are special features that can be incorporated into MD such as x-ray or photon bombardment, surface structures such as silicon or graphite, applied electronic current, radioactive elements, and others important in medicine or general biology. These are mentioned in most of the general software manuals with NAMD, LAMMPS and Gromacs providing great detail, but are usually utilized by engineers for such activities as Plasmon chip design, diagnostic components, or equipment in the medical fields [7,54,55]. Nevertheless, these constitute one area of focus related to biologics, especially in basic testing phases such as kinetics or expected interactions related to membranes.

Invariably in creating initial membrane models, researchers will encounter or wish to use lipids or other membrane components not found in the FF libraries for the respective software. Creating these initial models, termed topology for individual molecules FF, constitutes the second decision for proper MD experimental design. It is well worth the effort to make sure that these are correct, and several sites exist for initial FF library entry generation. The most commonly used sites are PRODRG and automated topology builder (ATB), the latter with down-loadable libraries of specialized lipid or membrane FFs [56,57,58]. Taking time to look at tabulated data from physical chemistry studies, such as angles, bond lengths and small local charges to make sure topology is correct, can save time in many later steps involved with model generation. These in some instances can be the cause of disputable findings from a simulation as the basis of all calculations performed. Straightforward examples are a missing charge on an atom important in the normal interaction process such as a head group on a lipid, improper geometry related to end groups, or a charge placed in the wrong atom on a hydrophobic molecule. Real parameters can be found through literature combing for the respective compound in physical chemistry publications, tabulated physical parameters and older collections of physical parameters such as CRC or MERK manuals prior to 1988 [59]. In some cases, it may be necessary to include non-standard components such as myristoylation or glycosylation which follow the same approach to FF entry generation. With some of these components, there may exist deposited FF files, as NAMD, Gromacs and ATB maintain user deposited FF files. This is also true for specialized compounds of interest such as cyclic peptides, or secondary modifications such as haptenated probes or ligands, although usually maintained on lab specific websites.

Once these are generated, the initial coordinates can be set up using embedded software functions, or in some cases auxiliary software. This latter can be useful for membrane protein studies in particular, as most MD software packages are able to handle smaller components such as lipids. Examples are again NAMD, Gromacs, or LAMMPS [14,44], but there are a number of laboratories and groups with their own specific program found online, or add on functions such as g_membed in Gromacs [60,61]. The general process of membrane model construction is shown in Figure 3 and follows a very methodical approach for a generalized larger workflow process. It is usually more advantageous to completely create the membrane before addition of macromolecular components. More complex examples include the necessity to refold portions of larger macromolecules and are not the focus here. These often entail entire studies, especially if there is no homology models to begin with. Of note, there is often a wealth of structural data available for proteins or macromolecules from non-image based experiments such as spin labels, proximity fluorophores, or related studies, giving localized information for particular amino acids or compounds. These often pose more challenges for model construction, sometimes taking a month or more, but can generate accurate models in lieu of refolding simulations. Also, these serve as methodologies for checking the proper protein structure before more advanced simulations can be conducted, or judging the accuracy of results from refolding simulations [62,63].

The third and final variables in the decision process for MD research are related to what the researcher wishes to observe, defining the simulation parameters. In many cases there is overlap between simulation types and what can be ascertained. Examples are energy and motion, which are intimately intertwined. In a normal affinity kinetic analysis, it is necessary to generate multiple runs from different conformational states and apply force to one component, called pulled or steered MD, for small or large molecules. Kinetics are determined by taking the mean Gibbs free energy changes from all runs, and include pulling the molecule of interest from one position in a bound interacting state to an unbound state over time. This also allows for the analysis of molecular conformational changes to be studied by default [12,14]. In most of these types of simulations, hybrid systems such as Gromacs with FF 54a7 will suffice [50]. More complex questions have arisen in recent literature related to the pharmaceutical industry. Direct permeability of membranes have been studied, from small molecules and peptides to larger protein complexes such as lytic proteins from T cells or the immune system [35,64]. These have posed questions not necessarily related to kinetics, rather, they test the modes of action or the overall process involved. Here, most of the researchers questions can be ascertained from extended all atom unrestrained simulations. These are often combined with a control membrane lacking the molecule of study, as controls become important, as in wet laboratory work. In some cases, these break down to kinetic based studies in the end, but are wide ranging in the literature overall as far as FF simulation types and setup, starting around 2000 [10,65]. More rapid simulations using coarse grained MD only generate relative kinetics, not close to those determined in laboratory experiments, but instead are used to test very general principles such as membrane association. Often these are combined with hybrid representations of a single component such as proteins to test very general aspects of interactions, while the more detailed atomic representations assess global changes.

Combined simulations are not limited to inclusion of QM in small regions of the simulation. With combined CG simulations specifically, these have taken two paths, with the second becoming more developed recently. The first type of combined model utilized more in past research represents specific components such as proteins, peptides or molecule of interest as all atom or partial atom representations. Typically, the membrane constituents remain CG representations. This process allows larger time steps, often in the microsecond range, to be studied, giving more accurate results for the component of focus related to association times. A second more modern use of combined sampling with CG simulations began in 2008 with Lindahl working with Gromacs [14], where all atom models were fitted to CG models at various time steps and analyzed as static structures. This has been furthered by others, especially in non-protein membrane systems, to now allow fitting of all atom models with adequate sampling space to ascertain a number of physical and chemical properties. Examples of this for non-membrane systems illustrate the process well, and show that structural and chemical pathways can be accurately modeled using the technique [66,67]. With larger models such as embedded proteins, however, there are often more conformational states. Smaller molecules shown in the referenced Nies or Ganeson have only 0–3 conformational states which can be fitted into the CG model space, akin to x-ray crystallographic density fitting. Macromolecules can include more than a million states, but only relevant areas such as enzymatic sites remain important, while the bulk are simply solvent based. This is particularly interesting for membrane systems, providing larger time frames for sampling with various processes such as insertion, folding or necessary lipid interactions.

More diverse examples of MD are emerging with significant increases in utilizing the technique for research. These include diagnostic chips, vesicles used as delivery systems, models of vesicles used in kinetic studies such as absorption-emissions spectroscopy, specific neurological systems involving two separate lipid bilayers, multiple bilayer systems such as in tuberculosis, and interesting novel membranes from fixed attached lipid filtration to Ag lipid coated nanoparticles [6,16,68]. Many more exist in the literature, with a general theme of more complexity in the past few years. Some of the largest simulations include membranes, such as the entire photosynthetic light harvesting system and light absorbing Rhodopsin from human rods in the visual system [33,69]. For MD in these areas, it is often more useful to use partial hybrid systems, with only non-polar hydrogen fused into the parent carbon atoms. Kinetics in these fit well and most of the design processes function exactly as the models indicate. For chip design and vesicle construction, this can be invaluable. In the latter, membrane stability under various conditions such as temperature and sheer stress is important, and very minimal changes to the membrane can have significant effects. A simple example of this is cholesterol, which affects membrane fluidity. A change from 1% to 5% weight to volume is the definition of lipid rafts versus normal membranes in human cells [29,30]. Extended models, based on size, have shown that these can be modeled well and cholesterol will localize to clustered regions. Analogously, the membranes themselves can be assembled testing different localized membrane component’s effects on the global membrane structure related to mechanical processes. These originally engineering studies can translate well to medical device studies, such as those on diagnostic chips or products that come into contact with human tissue, such as dialysis equipment. In the industry, there has been increased use of biological synthetic membranes, especially with diagnostic equipment contacting human fluids. Other simulation setup considerations vary significantly based on the software involved, and include choices for steered MD such as rate and force, options to fix atoms in the x-y or z directions, setting walls with various force parameters or, conversely, continuous systems where atoms move to adjacent sides of a defined unit cell when passing boundaries. There are choices for temperature algorithms, pressure, close contact force distances, and many controls for various force parameters, far too many to describe in detail in a single review. For these, the various manuals associated with MD explain the parameters in detail. It is well worth backtracking references from the original authors who designed these in any critique and design of good MD experiments. Together with the FF and model coordinates, these form the criteria for the entire MD, entailing one to two pages of instructions with set parameters for the software, usually termed the run parameters.

Once the model is constructed to the level of detail necessary for the experiment, the run parameters set, and the FF defined, the various MDs are run. This first entails equilibration of the system modeled before the actual experiment. Normally random velocities are assigned to each atom to mimic an initial temperature and pressure. Equilibration allows these to be normalized across the system. Usually this necessitates multiple MD where temperature and pressure are sequentially held as fixed, raising temperature until the final temperature and pressure are achieved. Then, the simulation is run for a period with no external applied forces to allow the natural configuration of atoms to be achieved. Additional aspects such as steric clashes, atoms placed too close and initial movements also occur during this step in the process. These produce extensive unnatural forces due to close range interactions which are removed by the process of equilibration. This is often important for membranes which have to adopt specific shapes based on overall charges in the system, and assume structures akin to the first MD which entailed phase separation [70]. If the membrane is constructed by hand, this process can take long time periods for membrane systems. Thus, there is a lack of starting files for more complex membranes available for general use, especially with vesicles, which can save time for other researchers. Still, many researchers start by utilizing fragments of publicly available structure files for membranes which have already been equilibrated to some extent.

## 3. Analysis of Membrane Molecular Dynamic Simulations

Analysis of MD encompasses a multitude of variables based on the types of simulations and methods employed. A wide range of various tools exist within the software used for simulations, and in many cases the overall trajectories generated by running MD can be analyzed across different software. External graphics software allows analysis such as Pymol, or this can be done with the internal visualization graphical user interface (GUI) if the software includes this [71]. One of the most commonly utilized is Virtual Molecular Dynamics (VMD), a GUI developed for NAMD, although Gromacs, CHARMM and other simulations can be visualized and analyzed [44,68]. Additionally, a number of tools exist for conversion of various formats of the simulation trajectories for cross analysis in different software, with a number of open source toolkits able to handle data from several different software packages [72,73,74,75]. Good examples of these are MDanalysis, MDtraj, PylipID, and growing consortia such as the VMD plugin library, scipy or PLUMED [76,77,78]. Often, researchers also utilize a number of personalized scripts for data manipulation which can be found through literature and website searches. It is also valuable to have some scripting knowledge for handling large data sets, data format conversion and generation of graphics. Most software for MD are also open source, meaning easy incorporation of user written code. Some of the most utilized aspects of analysis are mentioned here along with changes in trends, together with some common mistakes and pointers to resources for plotting analyzed data.

In MD, energy and motion are stored in trajectories, represented with the overall vectors stored for each atom involved in the simulation. However, these are meaningless in summed form, except for very rudimentary analysis, which often made up the bulk of larger simulations in the distant past. Very straightforward Gibbs and Enthalpy can be analyzed simply by comparing the summed totals of the entire system at an equilibrated starting point and ending point. As with other methods, these also require multiple simulations, but were utilized in the past for less time consumption. Here, CG simulations can also be utilized through fitted all atom models in a similar fashion. An an example, a membrane system with an embedded protein or molecule of interest can be simulated unrestrained at one point, and then again for a period with the system changed to a finalized state [79]. Energies are extrapolated by simply taking the difference of the two states. This was largely replaced with steered molecular dynamics as computers became more rapid and able to handle larger simulations. However in some cases, such as association rate studies or insertion rates of proteins, long time periods may be involved in the tens to hundreds of microsecond range. This makes CG back-sampling a newer valuable method emerging in the field of membrane simulations.

Pulled MD, synonymous with steered MD, takes simulations as a starting point and applies an external force to one or more components of the system causing them to move away from a fixed point [12,14]. This allows an entire plot for various energies to be computed, along with any changes in energies along an entire reaction pathway. These types of simulations also open the possibility for a multitude of more detailed analyses to be performed. In many cases, various components of the simulation, regions, surfaces, ions, specific membrane molecules or domains in a single protein can be separately analyzed. One common mistake is that initial MD set-up often requires individual atoms, groups or elements to have specific energies printed beforehand, in most cases to save individual file space, as simulations can generate extremely large files. This includes generating files to include in parameter files before simulations start; however, they can usually be extracted later from most simulations with auxiliary scripts. Direct energy profiles for elements of interest include total Gibbs, Enthalpy and Entropy. Usually, Entropy is inferred by determination of the two other energies, as it is more difficult to determine from direct analysis. Gibbs energy is usually analogous with the kinetics from wet laboratory measurements, which fit surface plasmon resonance (SPR) and isothermal titration calorimetry data (ITC) rather closely [25,80]. Like these two wet laboratory techniques, measuring enthalpy and free energy, most other kinetic factors can be inferred.

Other methods open to researchers for analysis of energy include principal component analysis (PC), direct vector analysis and correlations such as Pearson correlation coefficient or Pearson’s R values [81,82]. This latter is used for analysis of energies and motions, although any two components can be tested to see if they are correlated to each other. In membranes, this can be demonstrated by over a dozen experiments. For instance, it was found in some simulations that two of one lipid type will adhere to a cholesterol, while another 100 of the same type are prevented from interacting through hindrance by the first two. Thus, sets of cholesterol and the bound lipids will show correlations, but the non-bound ones will not. Use of PC is similar, and can also be used to test any two components of a system. In MD analysis, these are most often used to correlate the top three derived energy motion vectors as a means to separate various component’s contributions to the entire total energy. This is often advantageous, especially for systems incorporating proteins or larger molecules, and invaluable to any design process. For membranes, this can be used to analyze regions, specific molecules, rafts and sides of bilayers. Vector analysis for any extrapolated energies can also be analyzed individually for contributions from the breakdown into single Cartesian coordinates of x, y and z. This often allows the energy contribution from specific directions to be ascertained, such as motions of a single protein domain or molecule in the x or y plane, important in the analysis of various membrane systems. These can also be used to generate visualized trajectories of single direction energy or motion contributions for a molecule or groups of molecules, useful in analyzing specific features associated with single components of the model. Usually shown as exaggerating a particular motion energy, these help extensively with engineering, mechanical understanding, conceptualization and presentation.

Other analyses often used include radius of gyration, dihedral angle plots and root mean square fluctuation (RMSF). Until recently, these were the most commonly presented data, usually related to conformational changes in proteins. Again, these are often broken down into a Cartesian system, as in membrane analysis often only two relative directions become dominant. These are most often applied to individual molecules within the modeled system, rather than global analysis in a specific membrane. An example includes the use of these analyses in ceramides within a lipid membrane system, which cluster and move between layers of solvent exposed regions based on charge and concentration gradients [83]. With protein analysis, these are invaluable tools for determination of amino acids associated with binding sites, especially in membrane systems where specific lipids or cholesterols are necessary for the structural integrity [31,84,85]. All of these are usually plotted against time, but as with PC analysis can also be plotted as a correlation between any two variables to determine relative correlations. With compared changes between starting and ending models, component analysis can often highlight coordinated movement against randomized effects.

More simple but useful analysis include distances, relative position over time and general motion. This latter can often be used to determine a randomized motion versus changes associated with perturbing a specific modeled system in steered MD. Distances are by far the most reported data originally, dating to the 1960-70’s, usually presented as graphs over time, both in older and recent literature. With membranes, this can be invaluable, and often simply requires determination of motion as a geometric function, with a single molecules as centered reference. Randomized motions show up in analysis as almost perfectly sinusoidal with time components, while non randomized effects will show direct shifts in motion behavior when plotted against time [14,86]. Distances between two points of interest can be used in a similar fashion [87]. With membrane analysis, it is also often useful to plot various single components in two dimensions over a time course from the simulation. These can be used to show clustering effects, or other effects from randomized versus nonrandom motion upon perturbation. Examples include clustering energy and visualization of inositol lipids around proteins, and lipid raft condensation from cholesterol concentration. Some of the most common types of data display are shown in Figure 4.

More specialized analytical tools for membranes include their intrinsic ordered states. Simple parameters such as thickness globally over the space defined by the membrane occupancy of the unit cell, or more precise, down to the single lipid components, can be plotted. This allows localized effects from small molecules to larger proteins to be visualized over the membrane. In the same manner, density can be analyzed and surface charges on both sides of a membrane, usually all shown as surface plots or heat maps. More physical analysis such as applied voltage current effects and x-ray bombardment can also be incorporated, and visualized [88,89]. For radiation based simulations, there is often a need to represent a portion or whole run with QM MD. Fortunately there have been some recent advances with the implementation of DFT which greatly increases the feasibility of electron transfer based simulations. Analysis of these simulations shed light on enzyme based applications of biologics or small molecules, especially when covalent bond formation is important [33,47]. Kinetics from these simulations shift down to femtosecond rates and much higher repulsion and association energies. However, complete surface charges at the single atom level can also be computed and plotted, even giving a higher dimensional layer to analysis, down to electron energetic density maps at subatomic distances.

What is important for membrane based biologics and pharmaceutical development remains based on requirements of legislation. First, often kinetics are most important based on historically defied legislation, now including testing of protein mutants, designed Fab and whole antibodies effects with membranes. For non-lipid membrane interactions of different membrane types, this can be particularly important, such as polymer filtration or chemical based filters. These usually incorporate steered molecular dynamics, and for larger systems can be run in single precision on lab computers equipped with GPUs. Standard times include a single 24 h period for each MD run, with a benchmark of 200,000 atoms at 60–70 ns a day. This excludes the set up time for any system, but as a production of 50–60 unique runs for accuracy can be completed in 2–3 months [90,91]. For a larger corporation, or laboratories with a higher end computer, this can be reduced to less than a day excluding prior MD setup run time. In this case, the costs now allow pharmaceutical companies to integrate the use of MD as cost saving applications.

Secondly, direct interactions of products with their targets and off target effects being necessary. This can be particularly important for lipid based systems representative of the human cell membranes, as there are often clustering effects, and direct direct interaction of protein based biologics with non-desired membrane constituents both for targeted macromolecules and as distant effectors on other membrane proteins. For things representative of the neurological system, endocrine systems and specialized tissues utilizing membrane based signaling lipids, these can be decisive in effectiveness or off-target effects [34,92]. Standardized membrane models serve as a reference point to test new products safety and analysis therefore becomes standardized too. Often, diffusion rates and a single membrane component become a focal point, and can be difficult to localize and ascertain while still being present in the data. These models have been used in MD to analyze a more global interaction of various perturbations such as carbon nanotubes to the proteins involved in embedding into membranes such as Annexins [93,94]. Studies of this type can add validity to, or analyze various expected or observed health related issues. Also, they arre important in chip design, where lipid coated silica-dioxide or other compounds, ranging from gold to copper, are tested for proper interactions with ligands, ranging from small chemicals to large proteins [7,95].

Aside from these, a third set of important analytics in the design process are more product related for any membranes, coming from more specialized research. These often incorporate temperature effects, diffusion rates in two or three dimensions and concentration gradients between two enclosed systems [96]. Obviously this is important for studies of vesicle targeted membranes, where two membranes diffuse into a single bi-layer. Here, safety would dictate knowing the fate of all constituents of the vesicle. Other membrane systems such as polymer attached membrane filter interaction with molecules, proteins in fixed membranes, and specific small molecules interactions often incorporate the same analysis types [5,37]. In a similar manner, lipid membrane bi-layers have also been simulated to analyze ion gradient and solvent kinetic effects, with gated protein channels from neurochemicals to antibiotics, and even to pour forming peptides and compounds such as flavonoids. This is detailed by reading through the G protein coupled receptors (GPCR) database alone, which has ample examples of effects on the membrane, from lipid mediated cofactors to external small molecule drugs [97].

Current trends seem to follow an expected shift from MD used for small molecules in the past to macromolecules as the product. Several factors can be of interest in this manner of biologics development. Good examples include testing interactions from antibodies, or peptides with biological membranes. These are beginning to follow the more standardized approaches once used for small molecule membrane interactions. In the past and over the years, it has become routine to test the interactions of small molecules against several different membrane types, with a focus on different prostaglandins, leukotrienes or plasminogen ligands [98]. The effects studied are often these membrane molecules themselves, through direct or indirect interaction with the introduced molecules. Cellular membranes control a wide range of processes from vasoconstriction and dilation, uterus muscles and lining, muscle function, especially smooth muscle, gastrointestinal reactions, heart muscles, neurological reactions, inflammation, blood clotting and several other processes. These effects are controlled on minute to hour rates and involve membrane soluble and often embedded hydrophobic hormones. In the past, most protein biologics MD studies have only explored direct protein–protein or protein–ligand interactions, excluding these as constituents. As a number of pathologies are associated with the mentioned large number of membrane compounds, it can become standardized methodology to analyze a base set of biological membrane systems with anything targeting membrane components [35,99,100]. Invariably this can save money for market based research and development, by identifying potential side effects before moving to a physical product. This is especially important for membrane embedded systems, from more complex biologics such as CAR T cells, macromolecular ligands of membrane receptors, chimeric membrane receptors and antibody products targeting membranes [23,101,102]. Additionally, other membrane effects related to such things as carbon nanotubes, microplastics and medical device byproducts would benefit from use of standardized MD screening and analysis.

## 4. Experimental Design Principles Specialized for Biologics and Pharmaceutical Application

Using MD for research in the pharmaceutical industry has two main themes. There is either direct analysis of biologics as pharmaceuticals, or they are used for the design of equipment. A third theme is studying the actual methodologies themselves, which poses an interesting area of research often overlooked in debate. However, many of the most recent biologics ranging from CAR T cells or modified immune cells to proteins and peptides now include some level of MD in the overall patent process [20,53,103]. With appropriate methodologies, MD are utilized for scrutiny and design processes ranging from cell adhesion to surfaces and delivery system membrane interactions, to filter membrane interactions [4,104]. In addition to the scrutiny or design aid, MD can also be used to explain discrepancies between wet lab data, which are encountered with some products. A rule of thumb has been to check results against SPR, which tends to fit MD work more accurately than other techniques. What is needed for patenting and product design is based in large part on the specifics, which have an extensive range along with the current change in trends within the pharmaceutical industry. Some of these are explained below, but there is no exhaustive list as newer ideas and products arise.

Most basic principles of a designed biologic can be shown from proteins and chimeric proteins. Often, the mechanics of the product need to be ascertained in a proper environment, such as lipid membranes, acidic environments, or specific solvents expected during the products lifetime. These include domain changes and overall motions of the proteins, targeted or used, embedded in membrane systems. There is already a wealth of publications demonstrating these types of experiments, the most straight forward being GPCR, antibody targeted membrane proteins and ion gated channels such as KcsA [105,106,107]. Each of these demonstrates the effects associated usually with a secondary ligand or protein interaction such as small molecule pharmaceuticals, while dependent on the membrane itself for proper function. With GPCR, there is even a database now for different simulations that have been conducted [97]. In the simplest form, the GPCR is simulated in an unbound state, and then the interacting molecule or antibody is attached and simulated again, both in unrestrained simulation. Comparing the two allows binding affinities and overall energy changes to be calculated as the GPCR transitions into a different conformation. Like these, ion gated channels work in a similar manner; however, often with channels, the energetics of the ion gradients and solvent effects are also taken into account with multiple simulations at different pH, ion mixture or temperature. Unrestrained simulations can usually provide tabulated kinetics, though they lack specifics in the translational energy landscape, providing a more rapid result, including entropy without steered MD.

An interesting example of antibody targeting involves the T cell receptor (TCR) and accessible sites. In the initial blind generation of antibodies, two types were often encountered. One would cause constitutive activation of TCR and expansion of T cells. Another group would bind and label the TCRs while not activating them. While this seems trivial, this type of antibody targeting difference can be what determines overall biologics effectiveness. Obviously in such an extreme case, the first antibody would kill a patient, while the second would target specific TCR sub-types, eliminating a specific T cell. Thus one is a viable biologic utilizable as a beneficial treatment for T cell mediated diseases such as T cell lymphomas or Multiple Sclerosis. With the first wave of antibody biologics, this modeling aspect was not taken into account and in clinical trials the first anti CD20 killed participants by creating a cytokine storm linked to constitutive activation of the targeted receptor [108,109]. This occurred again for anti-CD28 and others until the overall structural effects of antibodies began to be incorporated into the process [110]. Related to antibodies, another key interest is the threshold for activity. It is counter intuitive that the highest affinity is not always the best. Often with immune receptors, there is a range, where too high an affinity can lead to autoimmune disorders [111,112]. This may be particularly relevant to biologics related to cancer, where there can be an over expression of a single protein, 2 to 50 times higher than that of other cells [113]. Affinity ranges can be used to target such cells, without causing large scale off targeting of healthy tissue in the design process. Here, precision kinetics can be invaluable through the use of MD experiments before actually producing any physical product. In these examples, the membrane plays a vital role in protein mechanics, accessibility to the target and through direct interaction with the targeting product effecting kinetics. Use of MD in these cases highlight the cost effectiveness, and foresight that can be achieved with the technique.

A similar type of study can be found for PD-1 affecting antibodies. Currently there are two types modeled and a number of generic biologics being developed with the same targets. In the broad scope, there are antibodies blocking the PD-1 receptor, which also have to be screened for activation determined by internal phosphorylation. The other group target the PD-1 ligand (PD-1L), thus constituting two receptors on antigen presenting cells [114]. In either case, the use of MD in simulations not only allows for binding sites to be studied, but effects on activity transmitted through either protein target [114,115]. Additionally, a wide range of products can be tested which is often pointed out as the downside of biologics related to patent protection and the pharmaceutical industry [116]. Current legislation has attempted to maintain the marketability of biologics of this type by requiring protein biologics to have greater than 20% unique sequence structure and to have to undergo the same 7–10 year process for patent approval and sales. Currently, generic biologics can only forgo this and move to rapid approval once the original product has been on the market for 10 years [117,118,119,120].

Other interesting principles include the effects of membrane constituents themselves as targets of biologics, and the direct effect of the targeting. Interestingly some of the earliest MDs included lipids such as leukotrienes or ceramides, in basic studies as drug targets, starting in the 1970s [1,121]. More recently, these have been used in several studies that demonstrate the range of possibilities from initial design and focus of MD for specialized purposes. Originally, cholesterol was used as a model study, starting in the 1980s [96]. Effects showed that cholesterol concentration highly regulates lipid membrane fluidity related to temperature. Studies then began to look at the effects of cholesterol in lipid raft systems and it became apparent that varied regions in larger membrane systems can have micro conformations based on localized concentration of components [122,123]. In these cases MD was used to further the theory of lipid rafts. In lipid rafts, the process has been modeled in MD and found to be controlled by cholesterol concentration differences creating less fluid structures. Not only are these “rafts” less fluid, but the membrane dimensions and charges change relative to the surrounding membrane they are embedded within. Later studies have now began to incorporate these lipid membrane systems into functional effects for a range of membrane proteins and related compounds of pharmaceutical interest [30,124]. In cells, this has shown clustering of surface proteins on membranes and cholesterol clustering, giving rise to clustering of other molecules, such as inositol-phosphate or prostaglandins. Often, the cholesterol concentration plays a role in the conformational or functional state of membrane receptors, which also effect targeting.

Three examples using lipid mediators can be found in simulations of direct antibody targeting, proteins involved using these as ligands, or use of specialized membrane based systems to target these compounds within a lipid membrane. Leukotrienes are numerous in the human body, functioning as mediators in everything from metabolism to inflammation, and are some of the major focuses related to pain and inflammation [125,126]. A first example in model MD design is to exam the targeting of these lipid based components by the antibodies, along with similar compounds. This has been done initially and can shed light on the field, based on antibody interactions with the cellular lipid membrane itself, an often neglected component of cellular membrane simulations [127]. A second example, effects of constituents and clustering of antibodies on the membrane surface targeting membrane proteins are simulated. Overall MD have shown what happens for therapeutic antibodies, such as novel cysteinyl containing leukotriene antibodies in development as therapeutics as they interact with the membrane as a secondary effect [128]. Making changes in lipid constituents such as ceramides, proteins or one of over 200 lipids that can be within a natural membrane, the information applicable to safety and feasibility can be tested before moving into cellular testing with a range of similar products. [29,100]. Often, clustering and concentration play vital roles in emerging pharmaceutically focused MD. Again using Leukotrienes, MDs have been used extensively to study the metabolic GPCR associated with their synthesis and metabolism. Much of the inflammatory process relies on rapid production and elimination of these from the membrane [107,129]. Thus, the GPCR and cytochromes associated with metabolism are often targets for pain and inflammation based therapeutics. A number of MDs have been conducted with these membrane bound proteins, including metabolically within the model cell membranes [130]. In a range of MD experiments, everything from leukotriene concentration to lipid types within the membrane and pH, as well as charge from the surrounding solvent or membrane components, have been modeled. Finally, more advanced applications of MD have even began to look at the targeting of leukotrienes with membranes from therapeutic delivery systems. These currently range from lipid and polymer coated gold nanoparticles to vesicles utilizing vesicle membrane embedded antibody chimeras [6,131,132,133].

Using all of the mentioned simulations as models, some underlying themes emerge for experimental design. Obviously, the researcher wishes to model the real world as closely as possible, but this can often be difficult with cell membranes, or synthetic membrane systems exposed to biological systems. A normal cell has hundreds of different lipid mediators, along with an equal number of proteins. In many cases, the membrane contains a multitude of cell specific components, such as eicosanoids. These in turn allow a cell to use the membrane as a control mechanism for extracellular signals which can effect distant cells in the organism, again through membrane mediators. Eicosanoids in particular allow a cell to control membrane proteins, or attract extracellular constituents from small molecules to macromolecular complexes. Thus it becomes important to identify what someone wishes to observe beforehand, as model membrane systems currently only utilize two to several lipids and possibly cholesterol. More recent published research has shown an increase in complexity of modeled membranes by one or two components such as ceramides or eicosanoids, along with one or two additional differing lipids. Originally, this was not a problem as most modeled vesicle membranes in attempts to correlate the kinetics derived from wet lab experiments using synthetic vesicles. Most of these wet lab protocols utilize one or two lipids only, such as PS, PC, DOPC, DPPC, cardiolipin or equivalent, and often neglect cholesterol, which is a major cellular component in human and animal cells. However, as questions turn to safety and function in living organisms, systems are increasingly becoming more complex. It is well worth any researcher’s time and effort to place emphasis on the starting model, as this is the limiting step in information obtainable from any MD. Additionally, the complexity of the membrane system in organisms can be a deciding factor in membrane associated products’ success. Interactions with specific leukotrienes, as an example, has hindered many product’s success or contributes to off target effects which give rise to adverse reactions.

Other weighted factor for MD experimental design seems to be “what does the researcher wish to observe?” It seems pointless to create an overly complex membrane system if the solvent is more important. Key factors can include kinetics, protein or biologic conformational changes, protein–protein, protein–lipid or protein membrane interactions, specificity, pH, solubility and global effects. These can be daunting in scope and it is often better to focus on specific elements that can be obtained from a specific MD experiment. Kinetics is straightforward, and usually used to compare various starting products, to test a specific unknown affinity or compare calculated affinities with wet laboratory determined kinetics [134]. These most often rely more on a ligand, but do include global membrane affinities in some cases, such as Annexins [135]. Here, simulations are usually a series of pulled, or steered MD, which provide mean kinetics along with several other factors such as temperature and viscosity [80,136]. Simulations like this also usually allow for more rapid analysis of conformational changes and account for half of association based MD [12,137]. Other association MDs, however. are more readily studied with only several extended unrestrained simulations for longer periods, from 400 nanoseconds to 5–10 microseconds [24,138,139]. Unrestrained longer simulations allow differences to be compared more effectively over medium time intervals. This usually allows extended interaction times and longer associations to be observed along with effects, especially for membrane system’s interaction with outside influences such as solvents, soluble molecules and soluble proteins. Additionally, these can show localized effects across larger membrane systems such as component congregation and clustering effects. Other factors can be tested by comparing extended simulations, with only a few replicates, and can be revealing in the function of uncharachterized proteins, membranes or molecules.

For the most part, slight variations of any of these main experimental designs would benefit any laboratory with either real life membrane goals or membranes used for targeting in the development of biologics and pharmaceuticals. A final aspect of the design of good MD is the types of detailed analysis the researcher wishes to conduct. Choices fall into many of the mentioned MD examples, but are defined in more precise detail based on the available techniques used by the software involved. Thus, a combination of software choice, force field and the overall MD itself all play together in the design process, with the observational goal driving the experimental design process. This often is extremely laboratory focused and usually entails some aspect outside of pre-established protocols ranging from a unique protein or new unmodeled compounds to the effects of ionized radicals. There is direct correlation to the interpretation expected from the design studies and the accuracy of the models involved, especially related to kinetics [140]. Thus another common theme is what level of accuracy is expected to mirror wet laboratory experiments. In most cases, models need only incorporate several of a class of the most common compounds as hybrid partial charges in a membrane to achieve levels within acceptable standard deviation P-values from other biochemical studies. Often even simplistic models, such as CG, will correspond to wet laboratory data within P-value ranges of < 0.05, but there tends to be better correlation to SPR and ITC determined kinetics with hybrid models such as 54a7, and even more with all atom [25]. For physics and electron transfer at the QM level, there are not very many MDs available with membranes to allow correlation statistics. In the few examples, there appears to be correlation with physical laboratory results in individual published work, with many arguments arising as to accuracy in reviews.

## 5. Applying Simulation Data to Product Development

Rational design approaches to any aspect of science dictate the utilization of computer modeling before advancing to creating and testing a physical product [12,21]. The overall aims are to reduce cost, but also increase product creation rate, product success rates and eliminate problems associated with side effects or unwanted reactions. As in many cases MD simulations coincide with various wet laboratory work, these then begin to serve as secondary checks to make sure designs work as expected. Because of the complexity of the organism, there will exist for some time unwanted and unpredictable reactions, but many of these can be predicted and removed with simple standardized models. Changes in computational speed, use of GPUs and cheaper memory mean individual laboratories can now utilize these in research projects. This then progressing MD research to a portion of a study rather than complete research careers in and of themselves. A once unheard of petaflop computer system can allow rapid MD pipelines for a single group, or even a decent two or four CPU two GPU computer can allow for introduction of more general studies into the laboratory. These cost from US $250,000 to US $8000 for the former and the later, respectively at the time of this publication, Figure 5.

Downsides to this approach come from expected areas when any increase in newer methods arise. Initially, a lack of sufficient researchers familiar with MD techniques meant a lack of peer review and direct criticism of simulations. Now, the main drawbacks are associated with a necessity to refold larger macromolecules without structural data such as X-ray or NMR. This is of course applicable solely to protein based components of any simulations. In many cases, there seems to be no adequate critiques of computationally refolded proteins, or adequate validation such as tertiary structure or Ramachandran outliers [141,142,143]. Automated refolding exists online now from protein data bases such as Swiss-prot and swiss-modeler, which have attempted to incorporate several scoring functions and checks for structural validity [144]. However, homology modeling, the core of Swiss modeling, requires at least part of the unknown protein to have homologous sequences to a known x-ray or other three dimensional structure [145,146]. Thus many resort to MD alone in protein refolding for model creation which can take extensive amounts of time, especially in lieu of any starting structured regions. Membrane association is often necessary for protein stability. This, however, remains largely unreported and underplayed in most publications, along with standardized quality control checks. The second dilemma is a lack of public access to the large data sets produced from simulations. Currently there are several attempts, and calls for a data base to upload raw data sets. With refolded proteins, a similar evolution appeared and there is now a repository for models, modbase, where the scientific community is allowed access to the structures utilized in publications and research [143,147].

Progressing into product development, the use of MD then becomes more akin to direct engineering work in the classical definition. Varied components of the membrane, proteins and small molecules become important with specific focal points. For membrane models, the components for delivery systems are moving towards a simple construction pipeline of a desired membrane model with the correct viscosity, integrity and embedded elements necessary for function [148]. This can be ascertained from lists or composite analysis of MD studies, using a library starting point. Usually only one single novel element is the tested variable in a library of starting membrane models for simulations. For proteins such as Fab used for targeting or more complex receptors, this is completely mechanical. With the use of engineering based classifications for regions and domains, anything from chimeras to completely engineered machines are possible [21,149]. An example is the use of 3–10 helices found in TCR and dozens of other proteins. Often these are used to forgo the normal gravitational effect needed for levers, applying force to one end of a platform with a fulcrum and a load further away. In particular, these are highly represented in membrane proteins, often utilizing the force connected to an α-helix as load opposite the 3–10 helix. With a pull on a single amino acid, the 3–10 helix turns into a normal helix, extending the associated α-helix, transmitting signals through the membrane. Analogously, membranes have rafts, areas with specific thicknesses based on constituents, inositol rich regions, protein rich regions, and localized effects from specific molecules such as glycolipids and ceramides that can be classified. While more complex, many areas can be fitted into base model classification schemes. Here the engineering aspect comes from sets of predetermined MD with the desired starting membrane models, used as an assembly line quality control to test novel products desired effects or for generation of synthetic membranes. Though similar, the engineering aspects related to membrane MD are more geared towards non-pharmaceutical products and research. 

Combined these trends indicate a slow move towards more automation in MD, which is usually the case with most scientific specialties. Examples are genetics and the progression from painstaking gel sequencing to next generation sequencers and, equivalently, computer sciences and programming itself. The rapidity which is now being achieved with MD means a much greater combination of cross-discipline focal areas such as genetics, toxicology and immunology, to name a few [96,150]. Using toxicology as an example, the use of predefined membrane models for metabolites, toxins and cellular byproducts analysis with MD means these can be tested rapidly by simply introducing the new compounds [22,24]. This is akin to the testing of novel compounds mentioned previously, where several cellular membranes of interest are designed and maintained as a base model system. These can be invaluable to medicine and medical studies for a range of designed products from macromolecular biologics to general compounds such as pain medications. Additionally, it allows for more global analysis of the effects rather than extremely difficult compound tracing in vivo, and for membranes this is particularly important as most cytochrome enzymes are membrane dependent [121,130,151]. These are the basis of small molecule alteration or digestion, as well as most lipid degradation or synthesis. Secondary compounds often create allergic diseases. A number of these have been modeled within membranes and published, with focus more on small molecule interactions.

Additionally, application of the analytical tools in a more regimented fashion means a higher rate of success in product development. Due to the nature of MD, especially more recently with less computational time, this adds more timely developed biologics and pharmaceutical products. Current trends in product development have been more focused on moving rapidly through cellular toxicity and into animal models. From some aspects this can be deceiving and most products fail in the first phases of human testing than any other, something costly in monetary and life terms [2,11,152]. Longer term effects and more human interactions can be ascertained even before moving into cellular testing. While something such as expression like CD19 being expressed on blood brain tissue can be tested by simply screening already composited NIH expression profile databases, direct interactions with other components of the membrane such as leukotrienes often have distant effects in an organism [98,153]. These types of interactions cannot be observed until the entire organism is exposed to the products and is often extremely species specific. Data mining is important to such work, with AI used for rapidly comparing or identifying a single lipid’s interactions across many different MD simulations.

Analogous to the engineering aspects, with a biologics designed interactions, many of the analytical tools discussed also show a standardized or pipeline based trend. Clear increases, in particular for energy vector PC analysis, in published work have emerged, now occurring in a large percentage of publications from 2019–2022. These, if conducted properly, can help in modular approaches to protein design, capsid design and vesicle design with the effects of membranes necessary to the overall functionality. For membranes in particular, these can be used to trace specific influences perturbing the membrane, down to single components, but are often limited to those interacting with proteins. In analyzing membranes directly with MD, current trends often focus on the simple global parameters such as density, surface charge, thickness and viscosity changes or extremely physical parameters such as heat capacity. In direct physical MD utilizing QM, a large amount of the focus is often ionization and effects on electron transfer through radical generation, or again with protein interactions. Radical formation is itself an important aspect of longer term damage and effects from usually smaller molecular pharmaceuticals, but is important to current chemotherapy and radiation treatments for cancer [89,154]. In adverse drug reactions, the majority of metabolites are processed through membrane bound cytochromes, changing into radicals, then forming covalent attachments with membrane components or proteins. With the growing number of cytochrome simulations, it would be easy to start incorporating MD data into AI based data pipelines.

Focus on synthetic membranes has also become more represented in current trends in pharmaceutical product design. Due to the nature of detection and sensitivity, membrane embedded proteins are one of the most useful tools in diagnostic chips for medical labs ranging from simple fluid analysis to phlebotomy [155,156]. In almost all of the chips, it is both beneficial and necessary to embed the proteins within lipid bilayer systems attached to Silicon, Ni, Cu, Ag, and Au platforms [157]. These are also used in chips other than SPR, such as assorted affinity assay chips for simple kinetic sensor devices, in a similar fashion [158,159]. Here, analysis of MD simulations is feasible and applicable to any patents, shedding light on any differences observed effecting accuracy, such as secondary interactions causing false positive or negative results. In the case of antibodies, such chips can also be used in research to rapidly detect off target effects such as from glycolipids, eicosanoids or soluble receptor ligands. In these cases it is often essential to identify interactions with membrane embedded targets, membrane lipids, proteins and other constituents before a product is moved into animal testing. The use of MD in the design process for membrane interaction modeling is akin to afamatrix chips and is progressing in a similar way, moving towards a larger number of automated tests per chip. Extremely useful standardized sets of modeled membranes for testing in silico are increasing in the public sector and are already implemented in industry in this manner.

Biological mimicking lipid membranes have also been utilized in novel applications ranging from artificial kidney design and various longer lived products outside of sensors alone. One focus found was the introduction of ion channels and use of MD in testing the effects of various external environments [160,161]. In many cases, the driving force for translocation across membranes is from external forces from ion gradients as the sole energy source [139,162]. This itself can also be demonstrated on free membrane simulations, with components such as ceramides moving to one side with charged gradients, or other unique lipid bilayer interaction. Here, CG membrane models are applicable to uses of synthetic membranes including aquaporins for water purification, synthetic endocrine type devices and filtration membranes incorporating covalently fixed lipids. In these cases, focus is often regarding the lifetime of various protein or other components, which need to be re-applied at specific points to maintain viability. For many of these instances, CG longer simulations are often utilized to test lifetimes of various added constituents [10,163]. However, there is now a clear move in research towards all-atom based simulations even for longer time course studies. These provide a basis for other effects necessary in device manufacture such as oxidation or detergent effects, with a number of product focused labs combining CG and other simulation types.

More advanced applications have also utilized these combined CG and all atom separate simulations to extracting PC analysis, electronic charged surfaces, and electron flow to model logic gates used in biologic circuits [68]. Here, surface analysis is used mostly for charge distributions at the sub-angstrom level surfaces along bilayers and PC analysis for electron flow, rather than to detect motion and energy contributions from protein amino acids or other molecules. Additional plotted parameters are added for polar or non-polar head groups of lipids, lipid or hydrophobic tail groups, localized charge densities, charge thickness and various parameters with a more engineering based focus. 

Another consortium, the human brain project has now begun to provide larger collections of MD trajectories for analysis [124]. This has now allowed AI research to begin incorporating MD. In several instances, complex membranes have been set up and a number of multicomponent protein systems embedded in these. Downstream proteins in signaling cascades have also been modeled into what have become large scale intracellular interaction cascades, based on the initial membrane embedded systems. The stated overall goal is to allow any researcher a set of base models for MD, to test any compound’s effects on the signaling process, with an obvious focus on ligand gated proteins and possible ligands. Extreme examples have shown changes that occur in simulations at the sub μs level, related to ion flow and kinetics correlated to changes at the minute and hour levels physiologically [10,35,164]. In this project, there is an interesting use of basic MD data analysis combined with artificial learning and correlation data from known physiological processes aided by artificial intelligence-like algorithms. The ultimate goal is simply the running of several simulations with a new ligand, biologic or small molecule and the resulting process at a larger organismal level becomes known. This project is still under construction, but benchmarks have been met and the use of data analysis from raw MD serves as a foundation.

## 6. Future Perspectives for Membrane Based Molecular Dynamics

Obviously there is now a growing utilization for MD in the entire patent application and product design aspects of the pharmaceutical industry. This is partially due to the change in the industry from small molecules to larger peptides, new delivery systems and protein based biologics [38,165,166]. Interestingly, the drive for using MD in larger corporations was already underway in 2000, with Hoffman La Roche and others hiring over a thousand research and development technicians for MD projects alone [167]. A number of biologics in the patent database, and undergoing the approval process, have already utilized several MD to study the products’ effects at the atomic level as outlined. The initial wave of these were often Fab, or whole antibody based, but are now growing and include everything from chimeric receptors expressed on cells to entire designed capsid like delivery systems [39,40,80,102]. Novel vesicle delivery systems have also utilized MD, and at least one has been approved for use as a nasal aerosol system. Additional examples include peptide antibiotics, cyclic modified peptides for cancer therapies, afamatrix chips, diagnostic chips and in some cases modified Au-lipid nanoparticles as delivery systems. All of these have utilized MD in the design process and included studies in safety as well as proof of the principle aspects associated with the patent process or applications themselves [55,168].

What does the future hold, based on current trends and changes in primary research and the pharmaceutical industry? As the price of conducting MD experiments drops, we have already observed an increase in overall application of this type of work. This translates to two separate aspects of research. In the industrial to academic setting, a move towards automation and standardization is invariable. This has already become evident from the GPCR databases’ inclusion of MD data, the human brain project and the use of several different membrane types in the examples presented. In addition, model databases and basic sets of modeled systems are being maintained in various repositories offering a range of initial membrane structures and components as already partially constructed tests or experimental starting points. Each of these allows a standardized beginning step to become incorporated, which is often necessary in industry as a point of reference for regulatory agencies to compare results. With research and development workflows, this means a standardized set of MD will be utilized in the near future. For small molecule systems and membranes, this is already becoming normal, where several lipid membrane systems are tested for effects of newer pharmaceuticals. With biologics, the trend is most likely to be followed with ever growing and more complex membranes as standards for direct targeting and to test off targeting effects which increase with larger molecular structures. This can greatly aid in safety issues before products are even created in physical form, during design phases.

The growth of newer pharmaceutical products such as CAR T cells, anti-cancer antibodies, interleukins or utilization of the immune system, in one way or another, through chimeric or designed proteins is also likely to follow a similar approach. This also changes with the advent of larger biologics to a greater generic or biosimilar market and invariably translates down to newer models of decision making from primary physicians [169]. Biosimilars themselves have already prompted changes to the approval legislation in the US and EU pharmaceutical system, altering what constitutes safety in the development process [117,119,165,170,171]. With standardized membrane models, new chimeric receptors in particular can be tested for stability and function before expression in cell systems. Here MD serves to eliminate excessive cost associated with a more randomized trial and error processes used in the past. As the majority of Fab or antibody products also target membrane bound ligands, their effects can also become more standardized, especially with databases of modeled membrane systems ready for each new product. In addition, with a standardized approach there becomes more credibility with regulatory agencies than with de novo membranes in MD with each product. This process most likely will apply to other products such as chip design, filter design and several other areas. However, other products such as specialized vesicle construction may not be able to follow standardization similar to normal vesicles. Most of the membranes involved utilize extremely novel systems such as metal attached or modified lipid based systems. Automation processes such as these are already applied to other sectors of the health industry, such as food and dietetics [121,172].

In the academic setting, MD is likely to become more feasible to most laboratories. As such, there will be a change, from laboratories whose sole focus was MD alone to the incorporation of other aspects of wet laboratory work as the number of laboratories conducting pure MD work will become minimal. Like X-ray crystallography, which saw a large move towards automation in the 1990s, in many laboratories MD will become just one of several tools utilized in research. This opens up many areas where some wet laboratory experiments are difficult for one reason or other and also as cross reference for functional theories developed through direct wet laboratory results. Along with the advancement of structural biology in general, this increase in MD use also changes the time course of research projects considerably. In earlier decades, an entire group could spend their whole career studying a single cancer. In current settings a single laboratory, with the correct tools, software and resources, could cure a single cancer every 4 years. In areas such as engineering and equipment design, the use of MD modeled membranes has already been utilized routinely for decades, starting with some of the first simulations used [28,70,172]. With an increase in use, there will invariably be more products related to electronics and use of membranes in complex equipment. An example is from rapid sequencing equipment under development, where MDs play a vital role. In the envisioned nanopore system it is possible to expand whole genome sequencing to minute time scales, with lipid coated pores [173]. In addition, membrane systems form the bases of human neuron and electronic interaction and allow for design principles to be tested before development of products, ranging from pacemakers to direct neural network connections. It is, however, impossible to determine what will arise with science as a whole, as often products or new technology are developed without foresight.

On the downside, with the increases observed in use of membrane simulations, there is also an increase in the need for peer review [174,175]. This is often more problematic with MD data, research and publications. First, the simulations themselves can generate large data sets, often tens to hundreds of gigabytes in size, while analysis can be equally cumbersome to deal with. Secondly, data related to the accuracy of any models, or simulation itself, is often not submitted or presented as part of the overall published data and again can be quite large. Simple tricks include looking at force field files from the simulations, starting models and the overall run parameters in the experiment [141,142]. In many cases, researchers are able to salvage useful information even from a badly designed experiment. An example is pulled simulations, where the unit cell parameters are too small, which is a common mistake for those learning MD. Often, motions involved are still significant and informative, even for small regions of respective membrane systems. Nevertheless, there are often unrealistic protein, small molecule and lipid structures presented in research, many neglecting simple structure-based knowledge from a century of studies. This has prompted some scholars and institutions to suggest or attempt to compile and maintain MD data repositories akin to the protein data bank or modbase [16,143]. In many cases, even deposition of the models themselves can be useful to the academic community, but are often prized by corporate research and thus protected outside the academic realm. Still, an investigator automatically maintains copyright from the date it was generated according to international law, although structural information in this manner has only been used in very few cases for intellectual property fights [117,118,120].

Utilization of tertiary structure knowledge and protein physiology in general can often aid in critiques, but does not often implement membrane systems. In many cases, logical answers for minor outliers exist which were unexpected, but global structural elements always maintain specific shapes, analogous to protein forms such as helices, sheets and well-structured turns. Similarly, models of complex membranes show common features such as rafts, charge localization and specific lipid–lipid type dominate clustering. For the most part, membranes can also be challenging to decipher differences from correct or incorrect models. With MD simulations, in most cases errors relate to FF files, or more specifically lack of proper equilibration, or even molecular elements. An example of simple physiological knowledge and error analysis is found in simulations of simple PS/PC lipid membranes used in many kinetic studies. When cholesterol is added, there are drastic changes in the melting point for the mixture. At 30% cholesterol, these become equivalent to butter at room temperature as do most membranes. In some cases, these were originally used for functional studies in MD, without connecting the two aspects, in order to study membrane binding proteins, thus effectively making the protein membrane interactions the same as those interacting with a rigid surface, rather than a fluid system.

Many of the grander aspirations of the academic community involve increased complexity of simulations. This is not simply related to number of elements contained in the models, but the size of simulation dimensions. We have already observed this, first with simulations of the light harvesting system, Mitochondrial ATP Synthase and Ribosomes without endoplasmic reticulum membranes, each simulation containing dozens of proteins, nucleic acids and membrane components or other macromolecular components [69,176,177]. Currently, the human brain project now maintains an MD database for simulations involving membrane proteins, membranes and related projects. Direct statements from this and other projects have already proposed the “whole cell simulation”, alongside complete modeling of the brain at all physiological levels. There is now increasing interest and attempts to further this project, where an entire human cell can be represented in an all atom MD. There have already been models and CG MD for some aspects of this, primarily from the Scripps Institute and associated labs across the globe [33,178,179]. Invariably there is a need for large scale membrane components, as the number of different membranes in a single cell include nuclear, golgi, phagosomes, proteasomes, lysosomes, excretory vesicles and a number of specialized elements for specific cell types. Some of these have already begun to be simulated, apparent from reviews of complex membrane systems, now including a number of intercellular membranes and bacterial cell membranes [180,181,182]. Additionally, each of these has localized membrane differences, such as lipid rafts and charge, or drastic pH and other solvent extremes on either side. Each usually also contain more than 200 or more lipid and lipid-like molecules distinct to membrane types and, invariably, differences between cells. Many regulatory hormones or other lipid components such as leukotrienes demonstrate this aspect, being synthesized in one cell type and having long range effects in the organism. Invariably, all these aspects will in the near future find their way into a model for simulations of the entire human cell. Expanded, these will hopefully encompass most or all cell and tissue types, incorporating cell–cell interactions which effect many membrane components. Tissue, often being constructed of more than one cell type and lined with different extracellular matrices, requires a large contribution from membrane based MD research when all facets are combined. Ultimately, the entire drive is to replace animal models with a testable, computer based organismal model of the entire human, a goal set for completion in the far future, the outcome of computer based automation of biological processes. Already this has found some funding drive from places such as the Swiss 3R Foundation, the US FDA, the EU Pharmaceutical regulatory legislation and, interestingly, animal rights groups such as PETA [183,184]. This something that would prove invaluable to medicine, pharmaceutical research, the pharmaceutical industry and the design of biologics in general. Current social goals would in the end allow any small or large molecule to be tested for any type of interactions in the human body with nothing other than a computer.

## Figures and Tables

**Figure 1 membranes-13-00148-f001:**
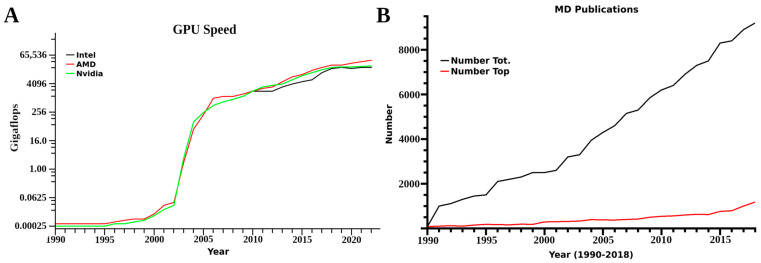
Change over time in computer speed and MD publications. (**A**) Change in speed for AMD, Nvidia and Intel Graphics processing units. Intel only began competing in the market in 2010 with non-embedded processors (black). Prior to the year 2000, Nvidia and AMD precursors companies are used, consolidated after 2000. (**B**) Increase in published MD work, with total publications (black) and top journals (red) compared. Data from authors own company price searches by year, literature searches by year, and ref. [1].

**Figure 2 membranes-13-00148-f002:**
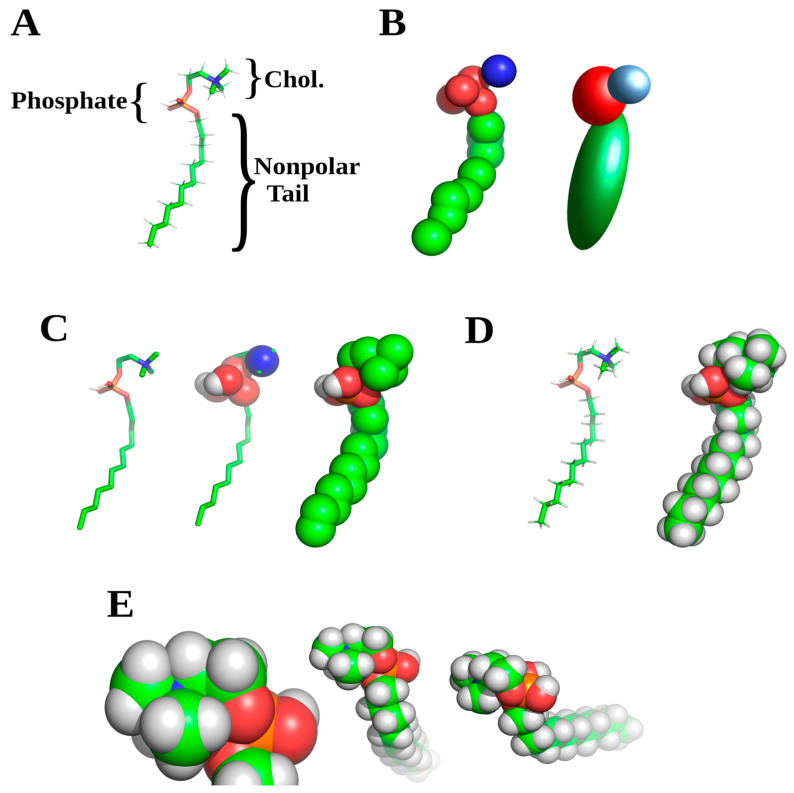
Force Field comparisons. (**A**) Di-phosphatidyl Choline (DPC) is shown with all atoms represented by sticks, Carbon (green), Oxygen (red), Nitrogen (blue), Phosphate (orange), and Hydrogen (white) (**B**) CG model, the atom radius space filling model, left, is represented by averaged space filling representations of charge. Neutral lipid tail (green) and head groups (blue, red) generate 3 space and charge models for the entire molecule. (**C**) Partial atom model. Left, only polar hydrogens are represented alone, while non-polar hydrogens are merged with the respective carbon, increasing point charge radius. Center, space filling model charge portions including hydrogens. Right, the final individual space filling and charge representations with merged hydrogen. (**D**) All atom model, left—stick representation, right—a final space filling model with hydrogens, inclusive of weak partial charges. (**E**) A close-up of all atom models with surface charges. Hybrid models burry these with larger radius composites. Small sub-angstrom charges shown as accessible surfaces, exposed colored areas of Nitrogen (blue) and Phosphate (orange), highlighted as green dots on the surface of the nitrogen in (**C**), middle, are particularly important for specific lipid interactions involving electron transfer, or lipids fixed to proteins or other molecules for extended periods.

**Figure 3 membranes-13-00148-f003:**
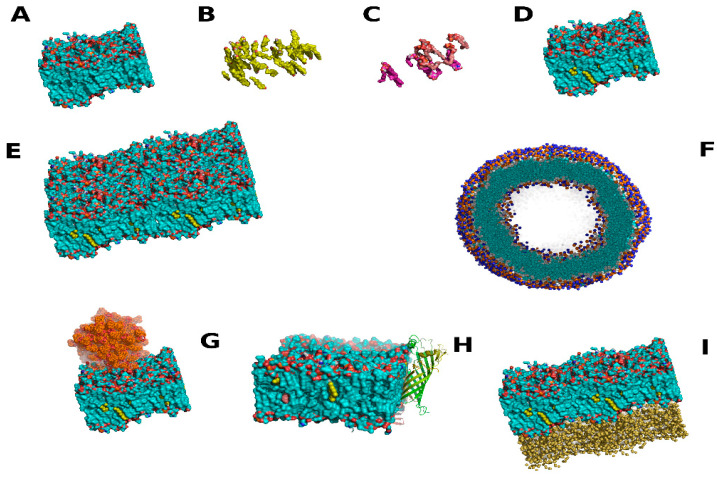
Creating initial structural models. (**A**) A starting piece of lipid monolayer is constructed by hand, or obtained. These usually have 1–2 lipid types such as DOPC or PS. (**B**–**D**), (**B**) cholesterol, yellow (**C**) Ceramides, purple, and Inositol Phosphates, Pink, are added to (**A**) using software to randomly embed these, or by hand with graphics software. This process is repeated until all membrane constituents are added, resulting in a small subunit (**D**) Final starting subunit containing several lipid or hydrophobic molecules. (**E**) The most common membrane simulation starting models are repeating duplications of (**D**), which can be generated with Pymol or other software to write out symmetry generated models, and combined. (**F**) More complex is a 5000 lipid vesicle used in studies, generated by software specialized for membrane vesicle generation, or (**G**) a hand created 60 lipid monolayer, orange, which has been extensively equilibrated before simulations. Other examples of the last stages after (**D**) are researcher specific. The most common (**H**) is a protein embedded or attached to the surface of a membrane. Software for this is presented in most MD suits, and these can be generated with graphics software. (**I**) A more specialized model for simulations, the bilayer, is attached to a solid surface, such as gold. This model again can be generated by graphics software, using software presented in suits such as LAMMPS, or with lab specific software sometimes shared online.

**Figure 4 membranes-13-00148-f004:**
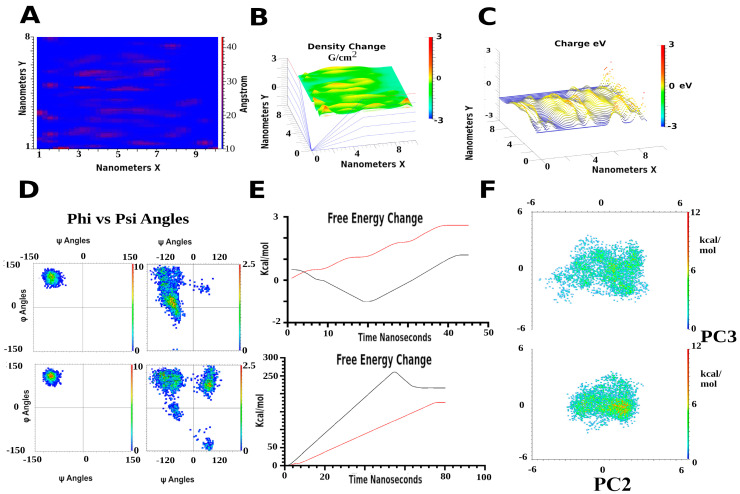
Common methods of data presentation. (**A**) Surface maps can be used to highlight differences based on a range of colors, including contour lines and changes between areas. (**B**) A 3D heat map graph, often used to highlight a range of data from density, thickness and charge to lipid clustering and type. (**C**) Another 3D graph type consisting of graduated lines often used to look at thickness and charge. In many cases, the graphs in (**A**–**C**) can be used to represent different data in a number of user-specific ways. (**D**) Phi-Psi dihedral angle change plots (Ramachandran) for a non-interacting, left, and interacting amino acid, right. The top and bottom are with and without the small molecule ligand. These graphs were often used in NMR, X-ray and earlier MD studies to identify important residues over proteins, especially in lieu of graphics software. The technique is also applicable to lipids, especially cofactors or ligands, but has been more often utilized recently by AI driven data mining. (**E**) Simple free energy change graphs for small molecules, top, and proteins, bottom. Proteins in some cases mirror the small molecules. In the top, red shows a multi-ordered affinity, which can be missed without pulled simulations when determining kinetics. The top, black, shows a small molecule which has a large energy barrier, even though the process is energetically favorable. The bottom, red, shows a very straightforward normal kinetic which has no hidden aspects, while the bottom, black, details a common aspect of membrane attracted proteins, an initial energy drop, before binding, due usually to hydrophobic residue interactions. (**F**) The most common representations, a PC plot of the energy motion eigenvectors for a protein, here binding a membrane. The top shows a membrane with 1% cholesterol, and the bottom 5%. The total energy for the protein interacting with the membrane is shown as the heat maps, highlighting more focused interactions from the protein’s bottom with 5% cholesterol. Data in (**A**,**B**,**C**,**E**) are test data for graphing; data in (**D**,**F**) are real experimental data (author).

**Figure 5 membranes-13-00148-f005:**
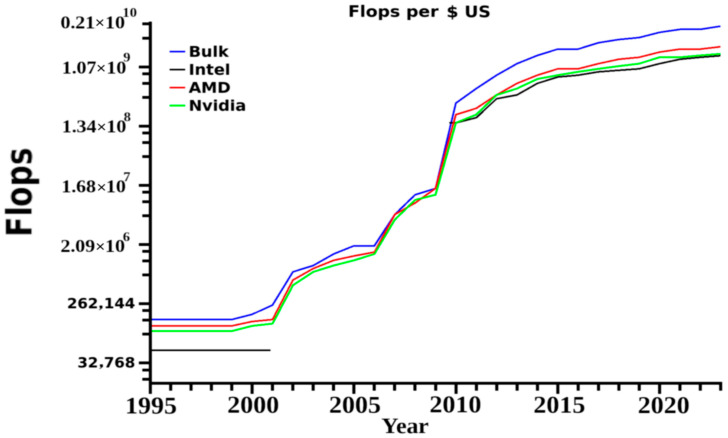
Number of flops per United States dollar by year. Almost mirroring the change in computer speed observed with GPU usage, the price per flop in computing power has decreased almost exponentially since early 2000. A recent plateau was observed around 2016, which is related to the limits of size in transistors. Intel (black) sold GPU’s primarily as PCI devices necessary to attach monitors from 1990–2000. In 2010, they began selling GPUs equivalent to Nvidia (green) and AMD (red). Currently Nvidia has exclusive purchase contracts with EU and US academic institutions, the US military as sole hardware supplier and Intel with the Chinese government and academia. This has prompted AMD to have lower prices in an attempt to sell GPU devices to the general public. Bulk (blue) are applicable to prices for all three companies when larger quantities of devices are purchased. A current standard GPU has a 20 teraflop (Nvidia, Intel) and 40 teraflop (AMD) processing capacity per single device.

## Data Availability

Data used in Figure 4 are from a complete study available on bioRxive at, http://doi.org/10.1101/2021.01.23.427770 (accessed on 14 December 2022), however are were not included in the final article.

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
