# Peer review of "Current Trends and Changes in Use of Membrane Molecular Dynamics Simulations within Academia and the Pharmaceutical Industry"

_membranes, 2023, doi:10.3390/membranes13020148_

Round 1

Reviewer 1 Report

The manuscript entitled” Current trends and changes in use of membrane molecular dynamics simulations within academia and the pharmaceutical industry is a detailed review focusing on the membrane molecular dynamics simulations. Although there are many reviews involving this topic, it still needs a review work to predict the recent trends covering the academia and pharmaceutical industry. This work is timely and suitable for publications on membranes. I mainly focus on the following aspects:

1.    The problem with the review is the way it is described. For example, there are too few illustrations in the review, which affects the readability. The author should add some graphs for the convenience of readers.

2.   In section 2, the author review several MD methods based on coarse-grained and all-atom models. The author had better give a detailed introduction to these methods, such as the fundamentals of these simulation methods and their basic processes.

Reviewer 2 Report

I like this article as it is written and the different points it covers. However, there is a lack of specific points, even if it is not in depth, on topics such as the use of web tools to build membranes and/or proteins, electrophysiology (some studies have been carried out studying different concentrations of salts, that is, creating a voltage difference between the two faces of the membrane) or the study of membranes with compositions simulating other membranes (plasma membrane, endosome, endoplasmic reticulum) or even membranes with different compositions in different monolayers.

The article is very dense and I think that some more figures should be included in which the structure and arrangement of lipids in the membrane could be observed.

Reviewer 3 Report

In the current review, the author has summarized current trends and changes using molecular dynamics simulation to solve problems within the academia and the pharmaceutical industry. The contents are solid and comprehensive, the paper is well-written, it basically covers the technics and methods that have been used in predicting and designing properties of membrane. But before the publication of the current review paper, some novel simulation methods, software, concepts and tools are worth to gain more attentions.

1.     In section 2, I think it is worth mentioning the development of the multiscale simulation technic (which offers the bridge to connect different scales, such as the coarse-graining for atomistic simulation to coarse-grained simulation and the reverse mapping or back mapping for recover the atomistic details from coarse-grained simulation), which has been used successfully in both biological and material areas. For example: the multiscale simulations of microphase separated polyelectrolytes by Venkat and co-authors (https://doi.org/10.1021/acs.macromol.1c00025, https://doi.org/10.1021/acsmacrolett.9b00478), the multiscale simulations of polymerization reactions by Erik Nies (https://doi.org/10.1002/adts.201800102, https://doi.org/10.1002/jcc.25348), top-down approach to study the self-assembly of peptide by Li et al, (https://doi.org/10.1021/acs.jctc.8b01025), and very recently the prediction of the structure and rheology of Galactomannan solutions by Juan de Pablo et al, (https://doi.org/10.1021/acs.macromol.2c01781). The coarse-graining and reverse mapping technics have been implemented in these studies, involving also the advanced simulation methods and useful software, such as the adaptive resolution method for the reverse mapping, the Espresso++ (http://www.espresso-pp.de) and the Votca (https://www.votca.org) software (the force matching method is widely used in biological systems).

2.     In section 3, I think some fancy tools (beyond the normal ones) that have been used widely and extensively for the analysis (post-processing) are worth mentioning, such as MDAnalysis (https://www.mdanalysis.org) and MDtraj (https://www.mdtraj.org/), they are almost compatible with lots of the MD simulation engines.

3.     For enhanced sampling, beside the traditional umbrella sampling and pulled sampling, the PLUMED package (https://www.nature.com/articles/s41592-019-0506-8) provides more advanced sampling method options, and it is also compatible with lots of major MD simulation engines.
